# The Effect of Residual Solvent in Carbon−Based Filler Reinforced Polymer Coating on the Curing Properties, Mechanical and Corrosive Behaviour

**DOI:** 10.3390/ma15103445

**Published:** 2022-05-11

**Authors:** Nurul Husna Othman, Mazli Mustapha, Nabihah Sallih, Azlan Ahmad, Faizal Mustapha, Mokhtar Che Ismail

**Affiliations:** 1Center for Corrosion Research, Universiti Teknologi PETRONAS, Seri Iskandar 32610, Malaysia; othmannurulhusna@gmail.com (N.H.O.); mokhtis@utp.edu.my (M.C.I.); 2Department of Mechanical Engineering, Universiti Teknologi PETRONAS, Seri Iskandar 32610, Malaysia; nabihah.sallih@utp.edu.my (N.S.); azlan.ahmad@utp.edu.my (A.A.); 3Department of Aerospace Engineering, Faculty of Engineering, Universiti Putra Malaysia, Serdang 43400, Malaysia; faizalms@upm.edu.my

**Keywords:** graphene oxide, epoxy, nanocomposite, coatings, acetone, EIS, ultrasonication

## Abstract

Solution mixing, which is one of the standard methods of producing Graphene−based Nanocomposites (GPC) may not be as efficient as it is expected due to the presence of residual solvent in the cured product. Therefore, the influence of including acetone in the preparation of Graphene Oxide−based epoxy coating (GO−EP) on the curing behaviour, mechanical and corrosive behaviour was studied. FTIR and TGA analysis confirmed that the GO−EP prepared by ultrasonication (GO−EP U) indicated the presence of more low−molecular−weight/low crosslinked (LMW/LC) sites than GO−EP prepared by stirring (GO−EP MS). Meanwhile, the tensile strength and hardness of GO−EP MS was 20% and 10% better than GO−EP U which confirmed that the presence of a lower number of LMW/LC could prevail over the agglomeration of GO sheets in the GO−EP MS. Pull−off adhesion tests also confirms that the presence of remaining acetone would cause the poor bonding between metal and coating in GO−EP U. This is reflected on the electrochemical impedance spectroscopy (EIS) results, where the GO−EP U failed to provide substantial barrier protection for carbon steel after 140 days of immersion in 3.5 wt% NaCl. Therefore, it is essential to consider the solvent effect when solvent is used in the preparation of a coating to prevent the premature failure of high−performance polymer coatings.

## 1. Introduction

The incorporation of graphene oxide in polymers are known to improve the barrier and mechanical properties of polymer coatings [1,2,3]. Graphene oxide has excellent corrosion resistance due to its high aspect ratio and ability to repel discrete molecules owing to their high potential energy barrier feature [4,5]. Therefore, the incorporation of graphene oxide in polymer coatings would improve the corrosion protection of metallic substrates. In a study done by Ramezanzadeh, B et al., amino−functionalized graphene oxide incorporated in epoxy by solution mixing method managed to increase the ionic resistance of the coating by forming numerous hydrophobic areas around the coating [6]. In another study, the passivation effect of GO managed to improve the protective performance of epoxy by forming tortuous pathways for electrolyte ingress [7]. Based on existing studies, the solution mixing method is a typical method used to disperse graphene sheets in the polymer via the ultrasonication method [8,9,10]. In this method, organic solvents are essential to ensure that graphene sheets exfoliate by allowing the solvents to act as a “molecular wedge” between the layers [11]. Once a homogeneous suspension is obtained, either epoxy or hardener is incorporated followed by further ultrasonication at moderate temperatures (typically 60 °C) to improve the dispersion and remove the solvent simultaneously [12,13,14]. Although these procedures are considered one of the standard methods to prepare graphene−based polymer nanocomposites, the problem might arise when solvents are not completely removed from the nanocomposite coating [1,5,15]. The alteration of the curing chemistry of polymers due to the presence of solvents during ultrasonication also influences the performance of the cured product.

Numerous researches have confirmed that most graphene−based polymer coatings fail due to the agglomeration of graphene sheets. However, none has highlighted that among the reasons for the coating failure was the change in the curing behaviour of the polymer as a result of the involvement of the solvent in the coating preparation method. Prominent studies have also confirmed that most of the common organic solvents are permanently adsorbed on the surface of the graphene oxide (GO) sheet [16]. For thermoset resins such as epoxy, solvents provide an important role in decreasing the resin’s viscosity during coating application. However, difficulties in removing them have already become a problem that affects the characteristics of the resin [17,18]. The encapsulation of the solvent within the resin matrix during the curing process may affect the final properties of the coating system [8]. A study done by Marcio et al. concluded that the presence of acetone in epoxy resins resulted in a decrease in the glass transition temperature because of the reduction in the degree of crosslinking and increase in molecular chain mobility [19]. Meanwhile, Hong and Wu et al. confirmed that the type of solvent remaining in epoxy samples affected the glass transition temperature (Tg), curing exotherm and reaction rate due to the absorption of heat during solvent evaporation [20]. Therefore, the change in network structure and curing behaviour is bound to affect the strength of epoxy. The entrapped *N*, *N*′−dimethylformamide (DMF) in epoxy provided a significant enhancement of mechanical properties [21]. Further analysis has shown that the decomposition of DMF at high temperatures during curing provided excess amine that improved the chain packing of epoxy [21]. In another study, the addition of xylene solvent into an epoxy system resulted in a softer and more flexible composite even after curing completes [22]. The interaction between epoxy and solvent resulted in a weaker three−dimensional network structure due to the reduced number of crosslinking. The presence of more voids at the surface and throughout the epoxy structure increases the mobility of large molecular chains resulting in a higher elongation at break [23].

Therefore, the goal of this study is to investigate the effect of including solvents in the coating preparation method on the corrosion and mechanical properties of the GO filled polymer coating. Acetone was selected as a solvent for this investigation as it is considered a universal solvent to facilitate the dispersion of graphene sheets during the solution mixing method [12,13,14]. To provide a critical discussion of this objective, a neat epoxy coating containing acetone was also analyzed. Several characterization methods such as FTIR, TGA and Electrochemical Impedance Spectroscopy measurements (EIS) were employed. The results are thoroughly discussed, and the conclusions drawn from the investigation were based on the analysis conducted.

## 2. Materials and Methods

The components for matrix resin used in this study consist of epoxy Resin EPON 828 based on diglycidyl ether of bisphenol A (DGEBA) with amine hardener F205 in ratios of 100:58 by weight (As advised by the supplier) provided by ASA Chemical, Malaysia. Graphenea, Spain supplied Graphene Oxide (GO) sheets in water dispersion. The GO sheets were initially extracted through vacuum filtration with a filter pore size of 0.45 μm, dried in an oven at 60 °C for two hours and weighed before being incorporated in epoxy resin. The lateral size of the sheets varies from 2 to 3 μm with a purity of 99.99%. Acetone (Sigma Aldrich, Selangor, Malaysia) with purity of 99% was used as a solvent for this study. The substrate used to apply the coating is carbon steel S50C, a common metal used for high−performance applications.

Neat epoxy pretreated with acetone was prepared by adding 20 g of DGEBA, 11.6 g of F205 and 5.05 g of acetone (16 wt%). They were added in a 250 mL beaker and were stirred vigorously for 1 h at 60 °C. Then, the blend was left in a vacuum oven for another 2 h at 60 °C until no bubbles came out. The mixture was then poured into a mould and was left at room temperature for 24 h and post cured in a vacuum oven at 100 °C. For comparison purposes, neat epoxy composites without acetone were also prepared by following the same procedure as mentioned above.

Meanwhile, epoxy−containing 0.19 wt% of graphene oxide was prepared through solution mixing and is designated as GO−EP U. This is by dispersing 1 g of GO sheets in 100 mL acetone by ultrasonication for 1 h until a homogeneous suspension was obtained. The suspension was then mixed with 177 g of F205 hardener and was further ultrasonicated for another 15 min before being placed on a hotplate with continuous stirring at 40 °C for 4 h. Then, the mixture was degassed in a vacuum oven for 2 h to remove the remaining acetone. 305 g of epoxy was then added, and vigorously stirred for 1 h at 60 °C. The blend was poured into a tensile dog bone mould with dimensions following ASTM D2370. A portion of the mixture was poured into a 5−centimetre diameter cylinder mould to prepare samples for the hardness tests. Another portion of the blend was applied using a paintbrush onto a sandblasted carbon steel plate for corrosion studies. The thickness of the coating was maintained at 215 μm measured by a Dry Film Thickness gauge. An epoxy mixture containing a similar weightage of GO sheets (0.19 wt%) without solvent (denoted as GO−EP MS) was prepared by following a similar method of preparing GO−EP−U, except that 1 g of GO sheets were directly incorporated into the hardener without acetone and were shear stirred for 1 h and ultrasonication for 15 min without heating. The samples were cured for 24 h at room temperature and post cured at 100 °C. To ensure the reproducibility of tests, each parameter was tested using more than two samples. Figure 1 shows the flowchart of the sample preparation process.

Fourier Transform Infrared Spectroscopy (FTIR Perkin−Elmer Spectrum One, PerkinElmer LAS Ltd., UK) was employed to study the changes in the functional groups of epoxy resin through different processing methods. The FTIR tests were conducted on the cured neat epoxy and GO−EP nanocomposites. Thermal gravimetric analysis (TGA) was performed on cured samples using a simultaneous DTA−TG apparatus with a heating rate of 10 K/min under a nitrogen purge rate of 35 mL/min. Using a highly sensitive ultra−micro balance, less than 5 mg weight of cured samples were weighed and placed in on a pan accurately. The samples were first preheated for 5 min to remove moisture in a ceramic furnace with a capacity of 250 μL. The furnace was then heated to 800 °C with a heating rate of 10 °C. Once the heating was completed, the furnace was cooled in less than 30 min under normal operation. The data during the heating process were captured by Pyris Software by providing curve optimization and calculations. All readings were repeated three times to ensure repeatability. To investigate the dispersion of GO sheets in the prepared coating, optical microscope images were taken using a Leica DMR microscope by capturing images from a glass slide that was dipped with a small amount of uncured coating. Using 100 W of halogen illumination, and automatic voltage adaptation and stabilization, the samples were zoomed up to 250× to capture the images of the hybrids at nanometric size. Transmission electron microscopy (TEM) was also conducted to study the dispersive behaviour of GO in epoxy at high magnification. TEM (Philips/FEI Tecnai F30, Laboratorio de Microscopias Avanzadas, Spain) was operated at 300 kV for ultramicrotome cut nanocomposite samples (dimensions: 6 mm × 15 mm × 1.5 mm) that are placed on the copper TEM grid.

To evaluate the mechanical properties of the samples, tensile tests were performed at room temperature with a crosshead speed of 0.5 mm/min using the Amsler 100 kN (Zwick/Roell, Inno Lab Engineering Sdn. Bhd., Malaysia) device equipped with 9600 series software. The samples for tensile tests were prepared by following ASTM D2370 and the test was replicated more than 5 times. Shore A hardness of the cured samples was measured using an indention hardness tester according to ASTM D2240. An indention was made in various areas of the composite with a minimum of 10 replicates for each sample. The average was calculated to obtain the final measurement of the hardness of the samples. The morphology of the fracture surface of GO−EP U and MS was observed by Field Emission Scanning Electron Microscope (FESEM) model JEOL JSM−6700F supplied by JEOL, Japan. The fractured samples were initially cut into small pieces to enable them to fit the sample holder. After that, the samples were subjected to gold sputtering to increase the conductivity of the samples to produce high−quality images.

The barrier performance of the samples was evaluated using the pull−off adhesion tests and Electrochemical Impedance Spectroscopy test (EIS). The coated specimens were subjected to pull−off adhesion tests to evaluate the adhesion strength of the coating system from the metal substrates. The procedure of pull−off testing using the adhesion tester Elcometer Model 108 was based on ASTM D4541. In general, the tests are performed by attaching a loading fixture (3 cm dolly) perpendicular to the surface of the coating and adhesive. After the adhesive is cured, the test apparatus will be attached to the loading fixture and aligned to apply force (tension) normal to the tested surface. EIS was conducted with a Metrohm Autolab electrochemical workstation device installed with Nova 2.1 software. Three−electrode arrangement cell was employed with a graphite electrode as the auxiliary electrode, silver/silver chloride (Ag/AgCl) as the reference electrode and coated carbon steel as the working electrode. The valid area of the working electrode is 1 cm^2^. The measurements were conducted after different immersion intervals of 1, 70 and 140 days in 3.5 wt% NaCl solution at a frequency range of 210 kHz to 10 MHz with a 10 mV amplitude. Finally, the EIS curve was fitted to an equivalent circuit using Nova 2.1 software to extract the Rct values.

## 3. Results

### 3.1. Characterization of the Curing Degree of GO−EP U and GO−EP MS Nanocomposite Coatings

FTIR analysis was performed to investigate the changes in the chemical bonds of GO−EP prepared by different processing methods. Figure 2a,b show the FTIR spectrum of GO−EP U and GO−EP MS. The difference in intensity of the characteristic bands especially around 917 cm^−1^ and 828 cm^−1^ (stretching vibrations of C−O and C−O−C in the epoxide group respectively marked in blue) indicate that the amount of uncured epoxy groups varies between samples [23]. GO−EP U contains a higher amount of uncured epoxy groups as denoted by the increased band intensity (Figure 2b) which is attributed to the influence of solvent in the curing kinetics of the epoxy resin caused by the interactions between reactants. Based on Figure 2a, it is evident that most of the functional groups are retained regardless of the participation of acetone in the processing of GO−EP nanocomposite. To imply that acetone is completely removed from the GO−EP U, all absorption peaks demonstrated in GO−EP U should be similar to GO−EP MS [19,23]. However, the FTIR results show that a sharp splitting of the imino group that is present in GO−EP MS at 1650 cm^−1^ is widened in GO−EP U [23,24]. The results suggest that the participation of acetone in the preparation of GO−EP nanocomposite coating reduces the formation of the imino groups in the cured layer. To justify the role of acetone in the curing behaviour of epoxy coatings, the FTIR curves for bare epoxy with and without acetone were obtained and displayed in Figure 3. As marked by the grey box in Figure 3, the bend at 1650 cm^−1^ for epoxy coating without acetone disappeared in an epoxy coating containing acetone (16 wt% EP). Therefore, the presence of acetone in the preparation of epoxy coatings was proven to hinder the formation of the imino groups during curing.

Therefore, the curing degree of both GO−EP U and GO−EP MS can be further validated by performing TGA analysis. The results of the report are depicted in Figure 4. The GO−EP U and GO−EP MS nanocomposite coatings undergo thermal degradation beginning at 124.98 °C and 126.2 °C with a mass loss of 1.5 wt% and 0.4 wt% respectively. The thermal degradation was followed by the maximum weight loss at onset temperatures of 336.7 °C and 351 °C for GO−EP U and GO−EP MS respectively. At this stage, the molecular chains began to decompose and the temperature to decompose 30 wt% of GO−EP U shifted to a lower region compared to GO−EP MS. This is ascribed to the presence of more unsaturated structures in GO−EP U, which had lower bond energies of aliphatic linkages that resulted in lower thermal stability. Hence, the processing method that requires GO to be ultrasonicated in acetone forms a nanocomposite coating that has a high number of low crosslinking areas due to the incomplete degree of curing. As demonstrated in the FTIR studies, the presence of acetone during the nanocomposite coating preparation hinders the formation of imino groups which lowers the curing kinetics of epoxy. This is consistent with the findings by Bilyeu et al. where the resistance of epoxy thermosets to thermal degradation is highly dependent on the degree of curing [25].

### 3.2. Characterization of the Dispersion Levels of GO−EP U and GO−EP MS Nanocomposites

The primary purpose of the ultrasonication of GO sheets in solvents before the incorporation in epoxy is to break the aggregates of GO into individual flakes. Previous works have also confirmed the effectiveness of ultrasonication in providing a good dispersion of GO sheets in polymers [9,10,26,27]. As remarked by the mentioned works, it is expected that the ultrasonication of GO in acetone would produce a highly dispersed GO−EP coating as displayed in Figure 5a. Although aggregates are present, their size is about 20% smaller than the GO sheets prepared by the stirring method (Figure 5b). Comparatively, the dispersion level of GO sheets in GO−EP MS is rather poor as clusters of GO agglomerates are found within the epoxy matrix. This implies that the GO sheets were not able to disperse homogeneously within the matrix. However, TEM images of GO−EP U in Figure 5c confirmed that the GO sheets become much shorter compared to the GO sheets in GO−EP MS in Figure 5d. This finding is consistent with other prominent studies, where ultrasonication would cause the reduction in the lateral size of the GO flakes due to the edge defects caused by ultrasonic shock waves [28,29,30]. Meanwhile, it can be seen that the stirring method retained the GO sheet size. This condition preserves the high potential energy barrier of the GO sheets which will reflect upon the excellent barrier properties of the final product [31].

### 3.3. Characterization of the Mechanical Properties of GO−EP U and GO−EP MS 

It is important to consider the mechanical properties of the GO−EP coatings as they determine how robust the coating would be when they are being exposed to the service environment. Hence, tensile tests were conducted to investigate the influence of the residual acetone on the behaviour of GO−EP nanocomposites. Figure 6 depicts the mean tensile strength of both GO−EP U and GO−EP MS. Based on Figure 6, GO−EP MS displayed the highest mean tensile strength with a value of 30 MPa. Meanwhile, GO−EP U obtained a mean tensile strength value of 20% lower than GO−EP MS (24 Mpa). The excellent tensile properties of GO−EP MS were observed due to the presence of more high molecular weight crosslinking within the nanocomposite matrix. Meanwhile, the hardness of GO−EP MS as listed in Table 1, is significantly improved when acetone was discarded from the coating preparation process. It is believed that the combination of the high−strength GO sheets and the highly cross−linking of epoxy matrix managed to provide ultimate mechanical strength. The results are consistent with the FTIR and TGA results, where the complete curing state of GO−EP MS enhances the interactions between the filler and matrix, resulting in high−quality reinforcements. On the other hand, GO−EP U showed a low hardness value (56 shore A) due to the low cross−linking state of the epoxy matrix.

Therefore, the microscopic analysis was conducted on the fractured specimens of GO−EP MS and GO−EP U that are depicted in Figure 7a,b respectively. Based on Figure 7a, clusters of GO sheets appeared on the fractured surface of GO−EP MS, indicating that shear stirring did not provide sufficient exfoliation of GO sheets during the preparation stage. Meanwhile, the GO sheets in Figure 7b exist as individual sheets with sizes ranging below 2 µm. This indicates that the ultrasonication of GO sheets in acetone during sample preparation played a vital role in ensuring that the GO sheets are fully exfoliated within the epoxy matrix. According to the theory of the matrix to filler interaction, small−sized particles would elevate the formation of strong interface layers between the matrix/filler that enhances the reinforcement of a composite [22]. However, this theory could not explain the results from the mechanical tests conducted in this study as the large−sized GO aggregates in GO−EP MS provided better reinforcement than the exfoliated GO sheets in GO−EP U. Therefore, this finding confirms that the tensile strength of the nanocomposite coatings is highly influenced by the curing state of the epoxy matrix. GO−EP U contains fewer high−molecular−weight chains which allow the cracks to initiate and propagate within the unsaturated areas of the coating which explains the low tensile strength of GO−EP U. Acetone not only reduces the saturated structures in the matrix by preventing the formation of imino functional groups but also increases the chance for the escaped acetone to form inclusions and pores within the epoxy matrix which become point defects for failure during mechanical tests.

### 3.4. Corrosion Protective Properties of GO−EP U and GO−EP MS 

#### 3.4.1. Pull−off Adhesion Test

The adhesion strength of the prepared coating was measured to investigate the effect of including solvent in the preparation of GO−EP composite coating. Figure 8 shows the adhesive failure of GO−EP U and GO−EP MS. GO−EP MS outperforms GO−EP U by displaying a typical cohesive failure with a locus fracture within a small area of the coating. In contrast, GO−EP nanocomposite coating prepared by ultrasonication GO sheets in acetone (GO−EP U) presented a typical adhesive failure on the majority of the areas covered by the dolly. This shows that the participation of acetone in the preparation method weakens the adhesive force of epoxy resin as indicated by the low adhesion strength measured by the pull−off adhesion tester. The adhesion strength declined significantly from 13 MPa to 4.9 MPa when acetone was included (Figure 9). Concerning this, the depressive effect on the adhesive force of the coatings is influenced by the combination of the effect of internal stress and the lower number of bonds on the metal/coating interface. The interference of solvent during the preparation of coating reduced the amount of chemical bonding such as imino groups that can be postulated by the lower degree of cure which is also observed in other studies [23]. In addition to that, the internal stress caused by the changes in the physical and chemical characteristics during the curing process potentially affects the state of the coating during the tests. Therefore, the coating might experience severe shrinkage due to the evaporation of the residual acetone during the curing process, thus affecting the metal/coating interfacial force. Meanwhile, the excellent adhesion of GO−EP MS is believed to be contributed by the complete degree of curing of the coating resin.

#### 3.4.2. Electrochemical Impedance Spectroscopy (EIS)

Electrochemical Impedance Spectroscopy (EIS) is a convenient way to study the effect of including acetone in the GO epoxy nanocomposite coating processing method. EIS measurements were conducted until the 140th day to observe the corrosion protection of the layer after prolonged exposure in 3.5 wt% NaCl solution. The changes in parameters based on a simulated equivalent circuit (Figure 10) for the EIS measurements are used to study the changes in structure and the corrosion resistance of coatings [28]. The impedance data enlisted in Table 2 are based on the fitting with the equivalent electrical circuits, which represent the physical processes taking place in the studied system. The terms R_s_, R_c_ and CPE_c_, are the solution resistance, coating resistance, and constant phase element of coating, respectively. Warburg constant (R_w_) is an additional property introduced to consider the conditions where the diffusion effects dominate corrosion [32]. To consider the non−ideal nature of the electrical elements in the coating system, the term coating phase element (CPE) is used as the coating capacitance in both the equivalent circuit.

As corrosion progresses, the diameter of the curve in the Nyquist plot shown in Figure 11a, d and g decreases for both GO−EP U and GO−EP MS from day 1 until 140 because of the substantial decline in the coating resistance. Based on Table 2, both GO−EP U and GO−EP MS initially had a very high coating resistance (R_c_) which is larger than 10^10^ Ω cm^−2^ during the first 70 days of immersion. This confirms the excellent barrier properties of the nanocomposite coating. The EIS measurements after 70 days revealed that GO−EP U outperformed GO−EP MS with values of R_c_ about 34% higher than GO−EP MS. The exfoliated state of GO sheets in the epoxy matrix can explain this result. As the immersion time elapsed, GO−EP U showed a single loop, followed by a ‘diffusion tail’ which is inclined exactly at 45° to the real axis (inset in Figure 11g). The loop from the high−frequency region refers to the intactness of the coating, which indicates that some areas of the coating layer are still intact. However, the R_c_ value of this sample during the EIS reading plummeted down to 50% from the initial immersion day from 2.5 × 10^10^ Ω cm^−2^ to 8.5 × 10^5^ Ω cm^−2^. Meanwhile, the following straight line in the low−frequency region is attributed to the electrolyte diffusion−limiting process (R_w_) [35]. At this stage, the appearance of R_w_ in the fitting suggests that corrosive ions have begun to diffuse through the conductive pathways formed upon prolonged exposure to water, as illustrated in Figure 10b. It should be noted that during this stage, electrochemical reactions between the electrolyte and substrate have not yet occurred. Even so, the decline in the value of phase angle of GO−EP U identified at 10 kHz after 140 days of immersion as shown in Figure 11h, suggests that the coating is starting to delaminate and the intactness has been largely reduced [36]. Therefore, the involvement of acetone in the preparation of GO−EP U coating introduced a high number of low molecular weight areas in the layer, that acted as hydrophilic sites to create an interconnected conductive pathway for electrolytes to penetrate. In contrast, the R_c_ of GO−EP MS throughout 140 days of immersion in 3.5 wt% NaCl maintained high impedance values higher than 10^7^ Ω cm^−2^ and are considered to have excellent protection against corrosion [37]. Although the GO sheets in GO−EP MS were not as exfoliated as the GO fillers in GO−EP U, the tightly packed and high crosslinking epoxy matrix with reduced low molecular weight chains was able to maintain the barrier protection of the coating for an extended period.

Another parameter that is considered the most useful parameter to monitor the coating performance is by analyzing the value of the low−frequency impedance modulus (|Z|_0.01Hz_) [32]. The values of |Z|_0.01Hz_ for both GO−EP U and GO−EP MS are extracted and plotted in Figure 12. Based on Figure 12, both coatings provided excellent corrosion resistance as the low−frequency impedance modulus (|Z|_0.01Hz_) maintained high even after the 70th day of immersion. This is ascribed to the GO fillers’ ability to provide tortuous pathways for corrosive ions to reach the substrate. It is worth noticing that the |Z|_0.01Hz_ of GO−EP MS was slightly reduced by 30% as compared to GO−EP U after 70 days immersed in 3.5 wt% NaCl solution. The excellent exfoliation and dispersion of GO sheets in GO−EP U elongates the diffusion path of the corrosive species. The coating damage of GO−EP U was unexpected as the coating initially outperformed GO−EP MS in terms of barrier protection as reflected by the |Z|_0.01Hz_ value. It is well known that the epoxy matrix consists of high−density segments that are separated by low−molecular−weight chains [38]. The low molecular weight chains in the coating would slowly degrade with further exposure to the electrolyte. Hence, a large number of low−molecular−weight areas as a result of the curing behaviour of GO−EP U will exist as channels between micelles that behave precisely like conductive pathways for easy access to water as illustrated in Figure 10. Therefore, this explains the sudden coating damage of GO−EP U after 140 days of immersion in the 3.5 wt% NaCl solution.

## 4. Conclusions

The effect of the dispersion method of GO sheets on the mechanical and corrosion properties of an epoxy coating was studied. FTIR results show that the ultrasonication of GO sheets in acetone altered the curing behaviour of epoxy as indicated by the absence of the imino functional groups in the cured coating. The tensile strength and hardness of the GO−EP prepared through ultrasonication in acetone were significantly lower by 20% and 10% respectively than GO−EP made without ultrasonication. Also, the GO−EP prepared by ultrasonication in acetone finally failed after 140 days of immersion in 3.5 wt% NaCl solution due to a significant number of low molecular cross−linking sites formed during the preparation of coating. Therefore, careful selection of solvents to conduct ultrasonication must be made to prevent pre−mature failure of the GO−EP coating. This might give an insight into the preparation of epoxy composites with graphene oxide fillers following the solvent processing route. On the other hand, green alternatives without wasting solvent in the preparation process of GO−EP must be further explored.

## Figures and Tables

**Figure 1 materials-15-03445-f001:**
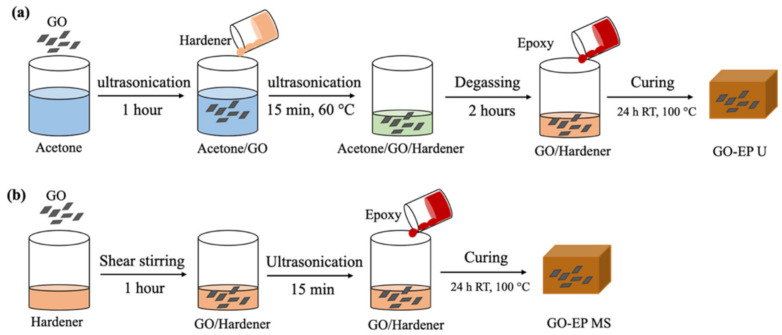
Flowchart for the fabrication of GO/EP nanocomposites through (**a**) ultrasonication (**b**) stirring. The samples for corrosion tests are applied onto a carbon steel plate which is not included in this illustration.

**Figure 2 materials-15-03445-f002:**
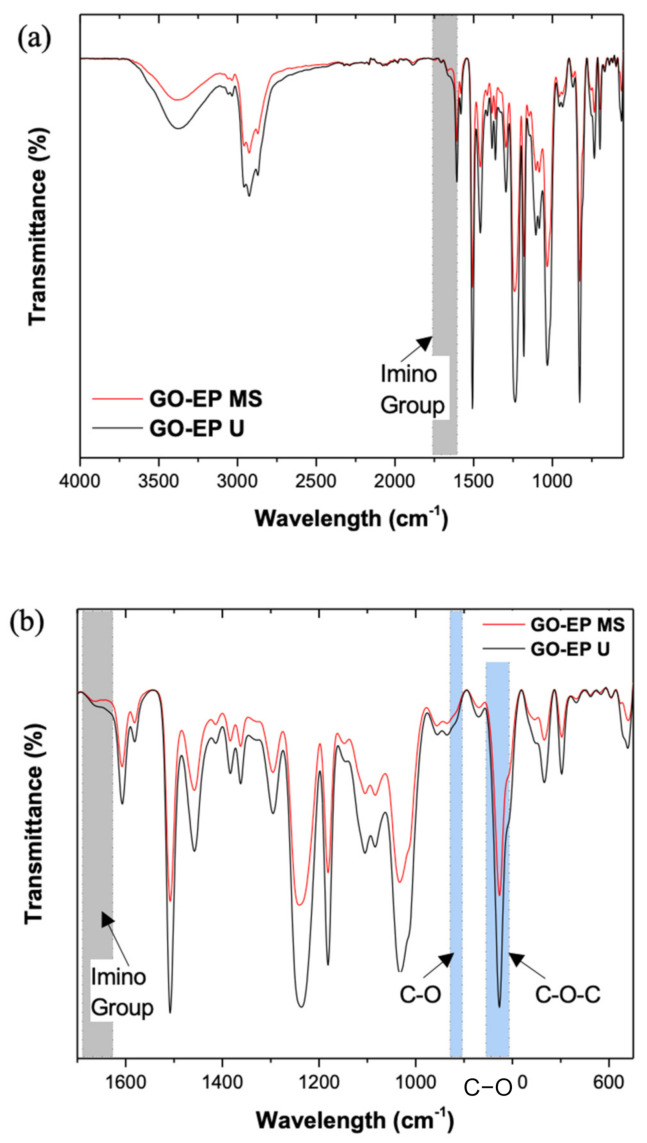
FTIR spectra of GO−EP U and GO−EP MS (**a**) Overall and (**b**) Zoomed in spectra.

**Figure 3 materials-15-03445-f003:**
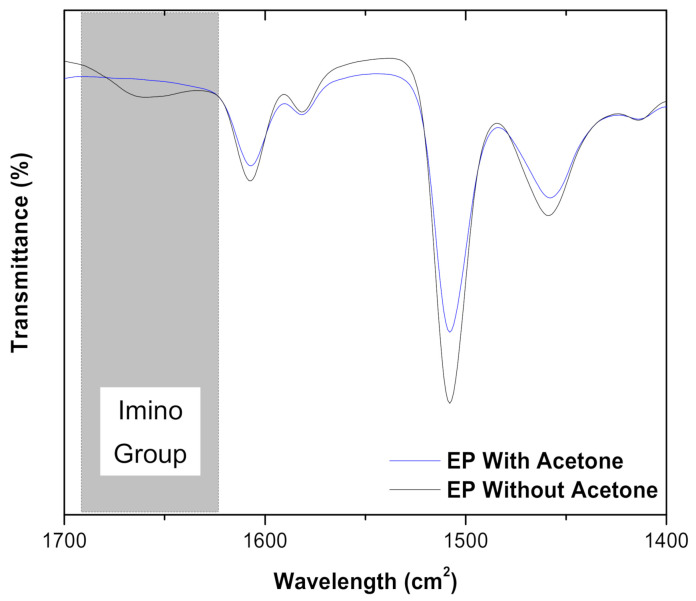
FTIR spectra of cured bare epoxy with and without acetone.

**Figure 4 materials-15-03445-f004:**
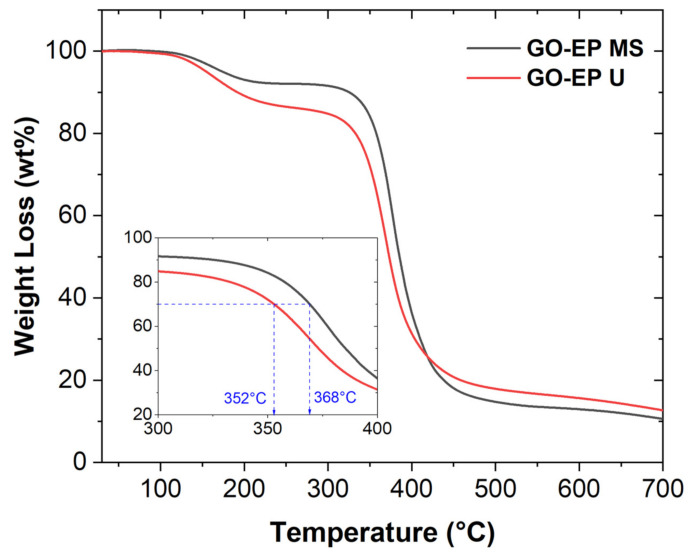
TGA curves of GO−EP MS and GO−EP U.

**Figure 5 materials-15-03445-f005:**
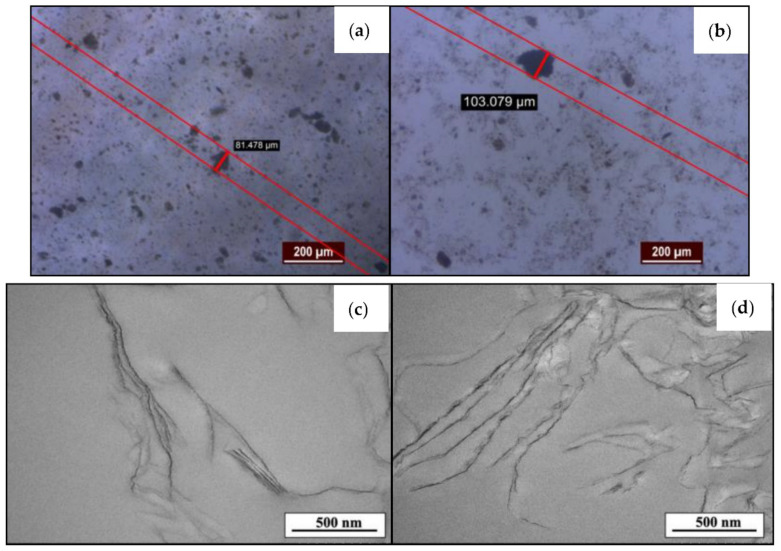
Optical microscopic images of (**a**) GO−EP U and (**b**) GO−EP MS. TEM images of (**c**) GO−EP U and (**d**) GO−EP MS.

**Figure 6 materials-15-03445-f006:**
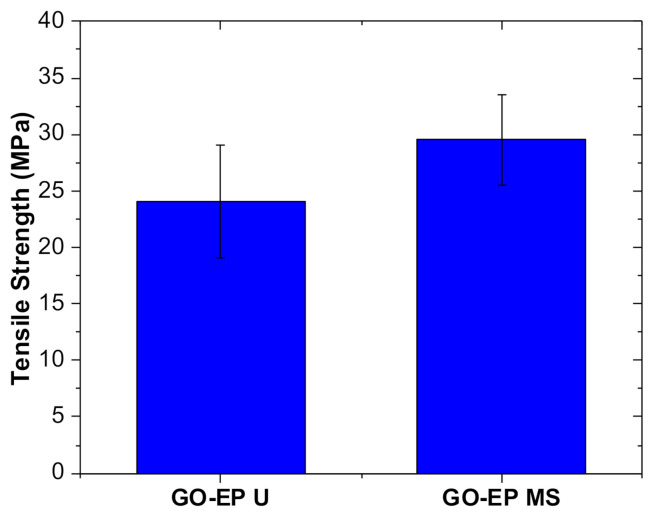
Mean tensile strength of GO−EP U and GO−EP MS.

**Figure 7 materials-15-03445-f007:**
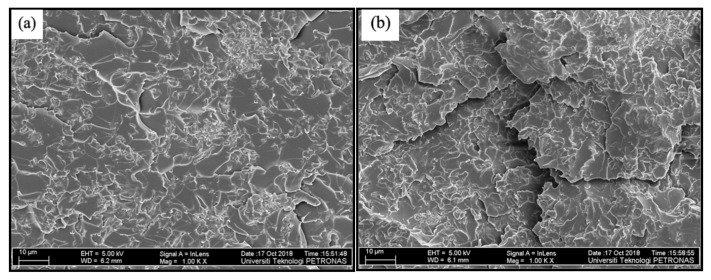
The fracture surface of (**a**) GO−EP MS and (**b**) GO−EP U.

**Figure 8 materials-15-03445-f008:**
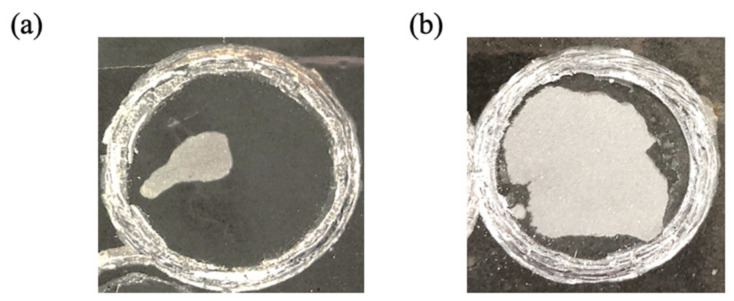
The coating failure after pull−off adhesion test on (**a**) GO−EP MS and (**b**) GO−EP U.

**Figure 9 materials-15-03445-f009:**
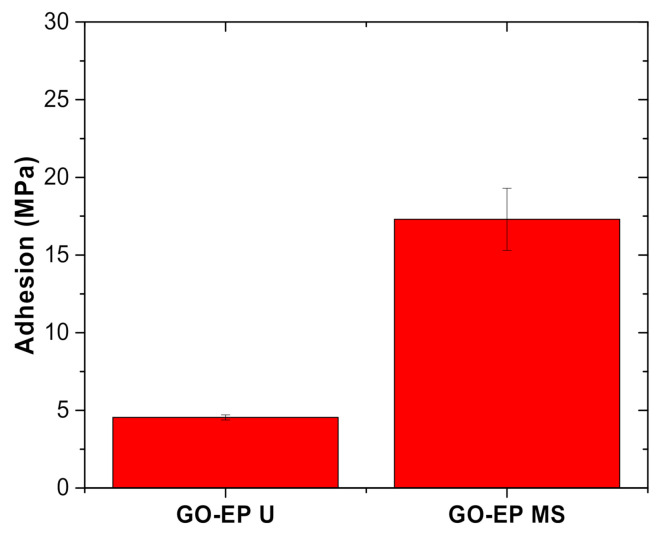
The adhesion strength of GO−EP U and GO−EP MS.

**Figure 10 materials-15-03445-f010:**
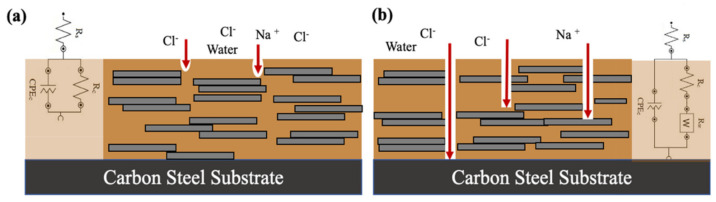
Schematic representation of the circuit model of three different scenarios (**a**) coating with high crosslinking areas (**b**) Coating that established conductive pathways through low cross−linking substrates that connect electrolyte to the substrate diffusion effects dominate corrosion) [33,34].

**Figure 11 materials-15-03445-f011:**
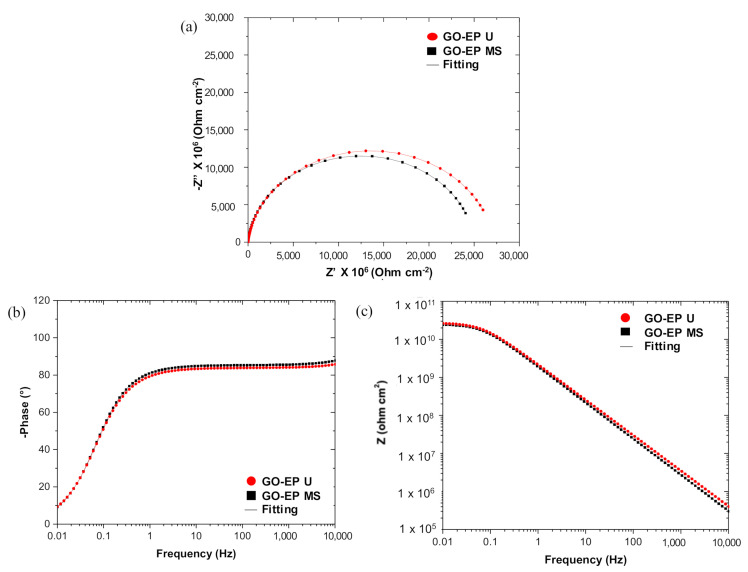
Nyquist and Bode Plots for GO−EP MS and GO−EP U at intervals of (**a**–**c**) 1 day; (**d**–**f**) 70 days; (**g**–**i**) 140 days.

**Figure 12 materials-15-03445-f012:**
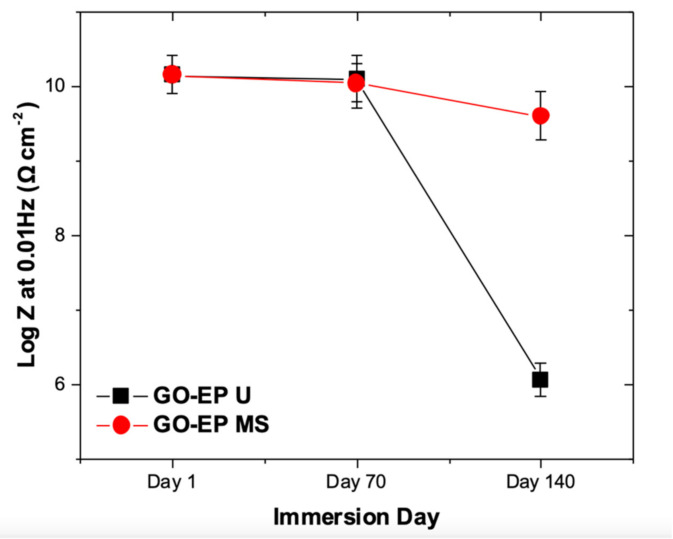
Compilation of |Z|_0.01Hz_ values of GO−EP U and GO−EP MS from the EIS measurements after immersion in 3.5 wt% NaCl solution for 1, 70 and 140 days.

**Table 1 materials-15-03445-t001:** Shore A hardness tests results including the standard deviation.

Sample	Shore A Hardness	Standard Deviation (%)
GO−EP U	56	7
GO−EP MS	63	6

**Table 2 materials-15-03445-t002:** Electrochemical EIS fitted values for GO−EP immersed in 3.5 wt% NaCl at different intervals. The standard deviation was kept below 5%.

Sample	Immersion Days	Rc (Ω cm^−2^)	CPE_c_		Wc (Ω cm^2^)
			Y0 (Ω^−1^ cm^−2 sn^)	n	
GO−EP U	1	2.5 × 10^10^	9.08 × 10^−11^	0.948	−
	70	2 × 10^10^	1.07 × 10^−10^	0.938	−
	140	8.5 × 10^5^	1.49 × 10^−10^	0.938	7 × 10^−6^
GO−EP MS	1	2.72 × 10^10^	8.22 × 10^−11^	0.932	−
	70	1.31 × 10^10^	5.14 × 10^−11^	0.948	−
	140	6.13 × 10^9^	6.28 × 10^−11^	0.928	−

## Data Availability

Not applicable.

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
