# Peer review of "The Effect of Residual Solvent in Carbon−Based Filler Reinforced Polymer Coating on the Curing Properties, Mechanical and Corrosive Behaviour"

_materials, 2022, doi:10.3390/ma15103445_

Round 1
Reviewer 1 Report
The dispersion method of carbon-based fillers on the curing properties, mechanical and corrosive behavior of filler-reinforced polymer coating was conducted. The preparation methods were explored and the related macro and micro properties were tested. However, the innovation of current research work needs to be highlighted, this is because there is already a lot of related research work to be carried out. The following specific comments should be considered to improve the quality of the paper.
- Firstly, the research background and application field of current polymer coatings should be emphasized in the initial abstract part. Secondly, there have been many studies on the addition of graphene oxide to epoxy resin. The authors should emphasize the innovation and significance of the current research.
- Introduction parts should firstly present the application fields and long service problems of thermosetting resin coatings. As known, polymers can only be used as anti-corrosion coating. However. in the long-term service environment, the coating may be degraded due to the long-term effect of temperature, humidity and loading, such as hydrolysis and plasticization, crack formation and propagation etc. By summarizing the main problems existing in the service process of resin coating, it is further put forward that it is very important to improve the mechanical properties and durability. For the performance degradation of resin in long-term service environment, please see the latest research work. Composite Structures, 2022. 281: 115060. Polymers. 2021, 13, 1902.
- Please provide some basic physical and thermodynamic performance parameters of epoxy resin and graphene oxide. What is the mass or volume ratio of graphene oxide in epoxy resin? Are there some variable parameters to analyze the effect of graphene oxide addition content on the properties of epoxy resin?
- Different types of performance tests should be distinguished, and the test details of each test should be provided, such as the required size, quantity and relevant test parameters.
- There seems to be no new peak appearing in the FTIR spectra in Figure 2 and 3, only the changes of peak height. What does the change of peak height explain?
- The thermogravimetric decomposition curve of TGA should give a full picture and a partial enlarged picture. Meanwhile, the quantitative discussion and analysis should also be given.
- For the dispersion levels of GO-EP U and GO-EP MS nanocomposites, transmission electron microscopy may be more meaningful than scanning electron microscopy. Please see this: Nanomaterials, 2021, 11, 1234.
- Why is there such a large standard deviation of tensile strength (Figure 6)? Is it due to the uneven dispersion of graphene oxide in epoxy resin? If the fillers are evenly dispersed in the epoxy resin, the failure mode of the sample is generally relatively uniform, and the local stress concentration is effectively avoided, resulting in a lower standard deviation.
- Figure 11 is too small. It is suggested to rearrange the layout and typesetting with high definition. In addition, all pictures should remove the redundant scales.
Author Response
Firstly, thank you for taking your precious time to review this manuscript. I hope that these answers and corrections could satisfy your comments so that our work could be considered for publication.

Reviewer 2 Report
The manuscript provides some information regarding epoxy-graphene oxide coatings prepared by two different preparation methods.
I did not witness a persuasive novelty and motivation behind this research. The field of epoxy-graphene derivatives has been at the centre of attention since almost 10 years ago and many studies have made thoroughgoing investigations in this field. Thus, any new investigation in this field requires innovative ideas. Therefore, my suggestion is to reject this manuscript.
Besides, the most problematic part of this research seems to be the preparation methods where GO is first added to the hardener. This will result in undesirable participation of GO in the reaction with hardener which will be followed by decreased reactive sites of hardener molecules that eventually hinders the efficient crosslinking reaction between epoxy and hardener. The authors should take this matter into account.
Author Response

(The authors gave the same response as above.)

Reviewer 3 Report
There are several comments and additions to the material presented by the authors:
- It remains unclear what the thickness of the coatings is and how uniform they are.
- It would be necessary to show the quantitative distribution of graphene oxide in the epoxy coating, as well as the actual distribution and orientation of the flakes along the thickness of the coating.
- It would be necessary to show how the development of corrosion processes occurs when the coating is damaged.
The comments made do not reduce the overall positive opinion about the work. Comments 2 and 3 may become the subject of further research, if they have not yet been conducted.
Author Response
Firstly, thank you for taking your precious time to review this manuscript. I hope that these answers and corrections could satisfy your comments so that our work could be considered for publication. Please see the attachment.

Reviewer 4 Report
Reaching the conclusion using 3 data points is difficult. It may be misleading (figure 11)
The language of the manuscript is very turse/complex.
Figure 11. please increase the fonts of X and Y-axes. One could barely see it.
In the caption of Figure 10, Figure 11 is mentioned.
Author Response
Firstly, thank you for taking your precious time to review this manuscript. I hope that these answers and corrections could satisfy your comments so that our work could be considered for publication.
At the beginning of this work, the objective was to address the disadvantages of solvent mixing, which is one of the common methods to disperse graphene in polymers. It is well noted that numerous ideas have been published, but from our point of view, there was a lack of work to address the disadvantages of including solvent in the processing method. Therefore, experimental work and discussions have been made towards this goal. That is why the common methods to process GO-epoxy based coating are used in this manuscript. Through this way, we are able to highlight that GO-EP produced by solution mixing has poor performance due to the presence of more low molecular weight chains as proven by the characterization tools used in this work. The decrease of reactive sites due to hardeners must be one of the reasons for the poor mechanical and barrier behaviour of the samples, however, it is relevant that the effect of residual solvent might also be one of the core reasons for this to occur.
Therefore, the title, abstract and introduction are modified to further convey the objective of this work.
Round 2
Reviewer 1 Report
The authors have tried to give a comprehensive response to the comments of the reviewers, there is still a problem that needs to be further improved. The current paper studies the effect of Residual Solvent in Carbon-Based Filler Reinforced Polymer Coating on the Curing Properties, Mechanical and Corrosive Behaviour. However, there was no summary on the corrosion properties of polymer coatings in the introduction. It is suggested that the authors add relevant summaries to match the current research work.
Author Response
Thank you for reviewing our work. We have tried our best to address all the comments and suggestions put forward. Please see the attachment. Have a nice day and stay safe.

Reviewer 2 Report
I did not witness a persuasive novelty and motivation behind this research. The field of epoxy-graphene derivatives has been at the centre of attention since almost 10 years ago and many studies have made thoroughgoing investigations in this field. Thus, any new investigation in this field requires innovative ideas. Therefore, my suggestion is to reject this manuscript.
Besides, the most problematic part of this research seems to be the preparation methods where GO is first added to the hardener. This will result in undesirable participation of GO in the reaction with hardener which will be followed by decreased reactive sites of hardener molecules that eventually hinders the efficient crosslinking reaction between epoxy and hardener. The authors should take this matter into account.
Author Response
Thank you very much for reviewing our manuscript. We have tried to address all the comments and suggestions. Please see the attachment.

Round 3
Reviewer 2 Report
The manuscript provides some information regarding epoxy-graphene oxide coatings prepared by two different preparation methods.
I did not witness a persuasive novelty and motivation behind this research. The field of epoxy-graphene derivatives has been at the centre of attention since almost 10 years ago and many studies have made thoroughgoing investigations in this field. Thus, any new investigation in this field requires innovative ideas. Therefore, my suggestion is to reject this manuscript.
Besides, the most problematic part of this research seems to be the preparation methods where GO is first added to the hardener. This will result in undesirable participation of GO in the reaction with hardener which will be followed by decreased reactive sites of hardener molecules that eventually hinders the efficient crosslinking reaction between epoxy and hardener. The authors should take this matter into account.